# Platelet-Rich plasma Injection Management for Ankle osteoarthritis study (PRIMA): protocol of a Dutch multicentre, stratified, block-randomised, double-blind, placebo-controlled trial

LDA Paget,[1,2] SMA Bierma-Zeinstra,[3,4] S Goedegebuure,[5,6] GMMJ Kerkhoffs,[1,2] R Krips,[7] M Maas,[6,8] MH Moen,[9,10] G Reurink,[5,6] SAS Stufkens,[1,2] RJ de Vos,[11] A Weir,[11,12] JL Tol[6,13]

For numbered affiliations see end of article.

**Correspondence to**
LDA Paget;
l.d.paget@amsterdamumc.nl

## ABSTRACT

**Introduction** Platelet-rich plasma (PRP) is a potentially efficacious treatment for ankle osteoarthritis (OA), but its use has not been examined in high-quality studies. Systematic reviews show that PRP injections significantly decrease pain and improve function in patients with knee OA. Ankle OA is more common than hip or knee OA in the young active population; with a prevalence of 3.4%. PRP injections in ankle OA are shown to be safe and improve quality of life over time, but no randomised controlled trial has been conducted. Our randomised controlled trial will evaluate the efficacy of PRP injections for symptom reduction and functional improvement, compared with placebo, in the treatment of ankle (talocrural) OA.

**Methods and analysis** We will conduct the Platelet-Rich plasma Injection Management for Ankle OA study: a multicentre, randomised, placebo-controlled trial. One hundred patients suffering from ankle OA will be randomised into two treatment groups: PRP injection or placebo (saline) injection. Both groups will receive two injections of PRP or placebo at an interval of 6 weeks. Primary outcome is the American Orthopaedic Foot and Ankle Society score at 26 weeks. Secondary outcomes determined at several follow-up moments up to 5 years, include Ankle Osteoarthritis Score, Foot and Ankle Outcome Score, pain subscale of (0–40), Visual Analogue Scale score (0–100), Ankle Activity Score (0–10), subjective patient satisfaction Short Form Health Survey-36, Global Attainment Scaling and the EuroQol-5 dimensions-3 levels utility score. A cost-effectiveness analysis will be performed at 1 year.

**Ethics and dissemination** The study is approved by the Medical Ethics Review Committee Amsterdam Medical Center, the Netherlands (ABR 2018–042, approved 23 July 2018) and registered in the Netherlands trial register (NTR7261). Results and new knowledge will be disseminated through the Dutch Arthritis Association (ReumaNederland), Dutch patient federation, conferences and published in a scientific peer-reviewed journal.

## Strengths and limitations of this study

► This is the first study to evaluate the efficacy and cost-effectiveness of ankle (talocrural) osteoarthritis (OA) in a multicentre, stratified, block-randomised, double-blind, placebo-controlled trial design. A positive study outcome will have a significant effect on widespread availability and nationwide implementation of an intra-articular injection treatment for ankle OA with platelet-rich plasma (PRP). A negative outcome (no effect of PRP), will prevent the widespread use of a non-efficacious treatment on patients.

► The study population consists only of patients with isolated ankle OA and no concomitant OA in other major joints of the lower extremities that negatively affects their daily activity level.

► The PRP used in this randomised controlled trial is an available PRP product that is used in clinical practice.

► There is a multitude of commercial PRP products available that differ in their production method and content, and different treatment regimens regarding dose, timing and number of injections are used in clinical practice. The generalisability to other PRP products and treatment regimens remains unknown.

► The composition of PRP will not be analysed; this decision was based on the fact that PRP preparation and composition has frequently been analysed and is typically not performed in clinical practice prior to injection.

**Trial registration number** NTR7261.

## BACKGROUND AND RATIONALE

Platelet-rich plasma (PRP) is a potentially efficacious treatment for ankle osteoarthritis (OA), but its use has not been examined in high-quality studies. Ankle OA is

more common than hip or knee OA in the young active population.[1] The incidence of symptomatic ankle OA is estimated to be 3.4% in the general adult population.[1] Patients with ankle OA have a quality of life and physical functioning comparable with hip OA, end-stage kidney disease or congestive heart failure.[2] These relatively young (predominantly female) patients have an increased risk for decreased work participation and family care. The available surgical intervention (arthrodesis) is associated with significant functional limitations. In contrast to hip and knee OA, where joint replacement is an excellent surgical alternative for severe cases, there is a clear need for effective non-surgical interventions in ankle OA.

PRP is defined as plasma containing a concentration of at least 1 000 000 platelets/μL.[3] Growth factors are stored in α-granules within platelets, and are released in a selective manner on activation. Growth factors released from the α-granules of platelets are assumed to provide the regenerative benefits of PRP.

Recent reviews concluded that in animal models PRP can diminish multiple inflammatory interleukin (IL)-1-mediated effects.[4] Due to this local anti-inflammatory response, PRP might have an indirect analgesic effect. The second suggested effect might be protection of the cartilage from destructive pro-inflammatory ILs. This is mediated due to an increased messenger RNA expression of proteoglycan core protein in the articular cartilage and decreased chondrocyte apoptosis.[4] Consequently, PRP could positively influence the collagen network of the cartilage.

Recent systematic reviews on the intra-articular injection therapy in ankle OA found a lack of evidence from high-quality studies to assist in clinical decision-making.[5 6] When compared with placebo injections, hyaluronic acid or corticosteroid injections, PRP injections might significantly decrease pain and improve function in patients with knee OA.[7–9] Given the clinical effect on pain reduction in knee OA and a good safety profile, PRP is a promising non-surgical therapy for ankle OA.[10 11] PRP might delay the irreversible surgical options like arthrodesis. No significant adverse events have been reported in any PRP trials regarding ankle OA, knee OA, acute hamstring injuries or Achilles tendinopathy.[8 12–17] Clinical studies on PRP in ankle OA are limited to a single report of 5 cases and a prospective case series of 20 subjects, which both showed safety and significant reduction of pain at a mean of 16 months and 12 weeks follow-up.[14 18] Well-designed prospective randomised controlled trials (RCT) have not yet been performed.

## METHODS AND DESIGN
### Objectives
We aim to determine the efficacy of PRP injections in the management of ankle OA by comparing two groups, both receiving two injections of either PRP or placebo solution. We hypothesise that PRP injections are efficacious for symptom reduction and functional improvement

compared with placebo in the treatment of ankle (talocrural) OA.

### Study design
The Platelet-Rich plasma Injection Management for Ankle OA (PRIMA) study is a multicentre, stratified, block-randomised, double-blind, placebo-controlled trial design will be conducted in order to compare two treatment groups: PRP versus placebo (saline). After the 26 weeks follow-up of the last patient in the study, the principal investigator and coordinating researcher will be unblinded only after the analysis of the primary outcome. A flow chart of the design and follow-up is shown in figure 1. The study included its first patient in August 2018 and aims to include the last patient by January 2021, consequently allowing analysis and then deblinding to commence after the last follow-up (26 weeks) of the last patient by July 2021.

### Patient involvement
#### Active patient involvement through reviewing the proposal (reviewer level)
Active involvement of patients during the design phase was performed by two patients with ankle OA. Both reviewed the grant proposal and provided feedback.

#### Execution phase: patients will actively contribute
Active patient involvement will be secured by invitation of at least two patients for the annual trial monitoring and evaluation meetings (informant level).

#### Analysis phase: patient will review and interpret the results (reviewer level)
Using the blinded codes of the randomisation groups and after breaking the randomisation code, patients (also active in the design phase) will be given the opportunity to interpret the results from a patient perspective.

#### Dissemination phase: patient-oriented approach (advisor level)
Results and new knowledge will be disseminated through the Dutch Arthritis Association (ReumaNederland), the Dutch patient federation, conferences and published in a scientific peer-reviewed journal. Dissemination activities will comprise a social media strategy with active patient involvement to share results with online patient communities and an events attendance strategy enabling patient participation in seminars, conferences and workshops.

### Study population
#### Population (base)
Patients with ankle OA in University Medical Centres, teaching hospitals, general hospitals and private specialist clinics will be informed about the study. In order to participate, patients must meet the eligibility criteria documented below.

### Inclusion criteria
1. Severity of ankle OA pain on a Visual Analogue Scale (VAS 0–100 mm) ≥40 mm during daily activities

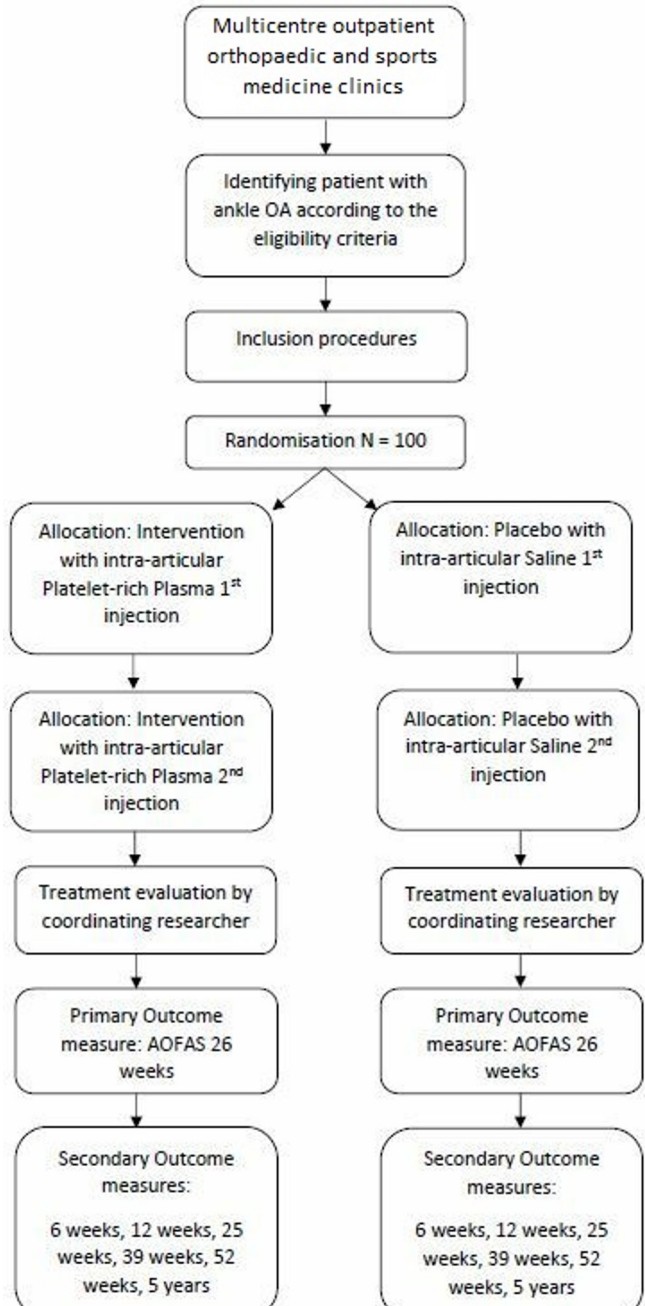

**Figure 1** Illustrating the Platelet-Rich plasma Injection Management for Ankle OA multicentre randomised controlled trial design and follow-up procedure. AOFAS, American Orthopaedic Foot and Ankle Society; OA, osteoarthritis.

2. X-rays (anteroposterior (AP) and lateral view) indicating ≥grade 2 talocrural OA on the van Dijk classification (clarified under the 'Radiographs' section).[19]
3. Age≥18 years.

### Exclusion criteria
1. Patient has received injection therapy for ankle OA in the previous 6 months.
2. Patient does not want to receive one of the two therapies.

3. Patient has clinical signs of concomitant OA of one or more other major joints of the lower extremities that negatively affects their daily activity level.
4. Previous ankle surgery for OA or osteochondral defects <1 year (not including surgery for an ankle fracture in the past).

### Radiographs
AP and lateral X-rays of the talocrural joints will be scored according to the van Dijk classification[19]:
1. Normal joint or subchondral sclerosis.
2. Osteophytes without joint space narrowing.
3. Joint space narrowing with or without osteophytes.
4. (Sub)total disappearance or deformation of the joint space.

### Randomisation, blinding and treatment allocation
We will include patients at the centre of their first outpatient clinic appointment. For each patient the coordinating researcher will prepare a syringe with PRP and a syringe with placebo (isotonic saline: 0.9% sodium chloride). A Good Clinical Practice (GCP) approved data management system (Castor EDC, based in Amsterdam, the Netherlands) will be used to perform a computer-generated block randomisation scheme with patients stratified to centre with a variable block size of 2, 4 or 6. This procedure will ensure treatment allocation concealment. The coordinating researcher, treating physician and patient all remain blinded to the allocated intervention. An independent researcher from the coordinating location will have access to the randomisation result and the allocated intervention. This will be relayed to a GCP-certified research assistant at the centre. The research assistant then selects one of the two syringes based on the allocated intervention and blinds the syringe with a covering sheath, ensuring concealment of the content of the syringe. The patients, treating physicians and coordinating researcher will all be blinded to the allocation of the intervention and to the contents of the syringe. The success of blinding will be assessed by asking patients which injection they think they have received just after the injection procedure, this will then be registered accordingly.

After the 26 weeks follow-up of the last patient in the study, the principal investigator and coordinating researcher will be unblinded only after the analysis of the primary outcome. The patients will remain blinded to the therapy until the 52 weeks follow-up (online supplementary appendix 1) of the last patient in the study.

### Intervention
In this study, patients will be randomised into two treatment groups: PRP injection or placebo saline injection. For each patient, the coordinating researcher will prepare a PRP and a placebo injection (isotonic saline: 0.9% sodium chloride). The PRP syringe will be prepared according to the instructions of the manufacturer.[12] Prior to commencement of the study, the coordinating

researcher was trained by a representative of Arthrex, as well as two experienced members of the PRIMA trial research group (GR and RJdV) with a vast scientific experience regarding PRP preparation and injection.[12 13] The PRP will be prepared using a widely used and commercially available system (Arthrex double syringe PRP system, Arthrex Medizinische Instrumente GmbH, Garching, Germany).

One syringe of 15 mL autologous blood will be collected twice from the cubital vein: at inclusion and at a time interval of 6 weeks. After blood collection the syringe will undergo 5 min of centrifugation and the injection will be given within 30 min following venepuncture to prevent formation of blood clots. No additional substances (calcium, thrombin or citrate) will be added to the PRP solution. For each procedure, 2 mL of PRP or placebo will be prepared and injected into the affected ankle joint under ultrasonographic guidance using a sterile technique. All patients will receive a second injection at 6 weeks, regardless of the effect of the first injection. To guarantee blinding for the allocated treatment of the patient, treatment assessor and treating physician, blood will be drawn and both PRP and placebo will be prepared according to exactly the same procedure for each patient (both at inclusion and at a time interval of 6 weeks after the first injection). However, in the control group, patients will be injected with 2 mL of physiological saline instead of PRP. An unblinded research assistant will blind one of the two identical syringes based on the allocated intervention using a specially manufactured covering sheath in order to conceal randomisation. Following the intra-articular injection, the syringe covered by the sheath (containing either the remnants of the PRP or saline) is handed to the unblinded research assistant, who will dispose of the syringe, therefore keeping the treating physician and coordinating researcher blinded. After the intervention, patients are advised to avoid heavy or repetitive stress to the ankle joint for a period of 48 hours. Furthermore, patients receive a leaflet containing usual-care healthy lifestyle advice beneficial for ankle OA. This includes losing weight and exercise (avoiding heavy or repetitive ankle loading exercises) such as walking, cycling or swimming.

### Use of co-intervention

Patients are instructed to avoid the use of co-interventions and non-steroidal anti-inflammatory drugs if possible, 24 hours prior to the intervention and if possible up to a year after the first injection. Throughout the study, key treatment (including usual care: exercise therapy and healthy lifestyle advice) and co-interventions used by patients will be registered, such as non-steroidal anti-inflammatory drugs, other analgesic drugs, intra-articular injections or inlays.

### Study procedures

In the event the patient meets the criteria for inclusion and exclusion, the patient will be informed in more detail about the study procedure. At that time, the patient can ask questions about the study and decide whether they will participate and sign the informed consent form. Subsequently, the patient will proceed to inclusion and the randomisation procedure.

### Inclusion

Patients are recruited for inclusion by their treating physicians at location. Following inclusion PRP will be prepared according to the PRP system instructions of the manufacturer. During the first two consultations, a total of two intra-articular injections will be documented with a time interval of 6 weeks. The patient will have no additional costs to usual care as a result of taking part in this study.

Physical examination will be performed on three occasions up to 26 weeks from baseline. Patient-reported outcome measures (PROMs) will be used to evaluate the treatment effect up to 5 years. A cost-effectivity analysis will be performed at 1 year using the PROductivity and DISease Questionnaire (PRODISQ). The time-frame of the follow-up questionnaires is further illustrated in table 1. Questionnaires will be managed and distributed digitally using a GCP-approved data management system (Castor EDC).

### Replacement of individual patients after withdrawal

In the sample size calculation, we compensated for an expected loss of 10% of patients to follow-up. No patients will be replaced after withdrawal.

### Safety

All adverse events reported spontaneously by the patient or observed by the investigator or his staff will be recorded. A variety of conditions have been treated with PRP ranging from muscle and tendon injuries to intra-articular injections of the knee and ankle. To date, no serious adverse events have been documented in the literature, concerning PRP intra-articular injections of the ankle. In accordance with the Central Committee on Research Involving Human Subjects guidelines, this study was classified as low risk for adverse events. Therefore, the local Medical Ethical Commission will be notified of any serious adverse events. In the event this happens, the advice of the Medical Ethical Commission will be followed accordingly.

### Outcome measures
#### Primary study parameter/end point

The primary objective of this study will be to quantify pain or functional improvement using the American Orthopaedic Foot and Ankle Society (AOFAS) score at 26 weeks follow-up. Studies evaluating the efficacy of PRP in knee OA maintained a follow-up between 3 and 12 months. We therefore opted to take 26 weeks for our primary outcome measure.[10] The AOFAS is a validated scale for ankle OA (0–100) measuring three subdomains (pain, function and alignment), which together total nine items.[20–23] The subdomain of pain is measured by

**Table 1** Time-frame of the follow-up questionnaires

| Follow-up | |
|---|---|
| Baseline | ▶ First intervention injections<br>▶ Physical examination<br>▶ AOFAS<br>▶ PROMs<br>▶ PRODISQ cost-effectivity |
| 6 weeks | ▶ Second intervention injection<br>▶ Physical examination<br>▶ AOFAS<br>▶ PROMs |
| 12 weeks | ▶ AOFAS<br>▶ PROMs<br>▶ PRODISQ cost-effectivity |
| 26 weeks | ▶ AOFAS<br>▶ PROMs<br>▶ PRODISQ cost-effectivity |
| 39 weeks | ▶ PRODISQ cost-effectivity |
| 52 weeks | ▶ AOFAS<br>▶ PROMs<br>▶ PRODISQ cost-effectivity |
| 5 years | ▶ AOFAS<br>▶ PROMs |

In addition to the AOFAS score, the following PROMs will be taken: FAOS, AOS, VAS, AAS, SF-36, GAS, EQ-5D-3L. These PROMs will be elaborated on further on. Furthermore, PRODISQ will be used to perform a cost-effectivity analysis. These questionnaires can be found in online supplementary appendix 1.
AAS, Ankle Activity Score; AOFAS, American Orthopaedic Foot and Ankle Society; AOS, Ankle Osteoarthritis Score; EQ-5D-3L, EuroQol-5 dimensions-3 levels; FAOS, Foot and Ankle Outcome Score; GAS, Global Attainment Scaling; PRODISQ, PROductivity and DISease Questionnaire; PROM, patient-reported outcome measures; SF-36, subjective patient satisfaction Short Form Health Survey-36; VAS, Visual Analogue Scale.

one item where a maximal score of 40 indicates no pain. Function consists of 7 items where full function is indicated by the maximal score of 50 points. Similar to the pain subdomain, alignment has a potential maximum score of 10 points using one item, indicating good alignment.[20 21] The AOFAS questionnaire, having undergone forward and backward translation to Dutch by de Boer *et al*, has an excellent internal consistency (Cronbach's α=0.947) and an excellent test–retest reliability (Intraclaas Correlation Coefficient 0.93).[20]

### Secondary study parameters/end points

Secondary outcome measures are a number of other PROMs. Specific time points of the secondary outcome measures can be found in table 1.

1. Ankle Osteoarthritis Score (AOS) is a Visual Analogue Scale (VAS) from 0 to 100 mm with 18 questions; 9 relating to pain and 9 relating to disability.
2. Foot and Ankle Outcome Score (FAOS): each question is assigned 0–4 points based on the answer given. The scale runs from 0 (extreme symptoms) to 100 points (no symptoms).

3. In order to evaluate pain, the pain subscale of AOFAS (0–40 points) will be analysed. On this scale, the lower the score the more pain the patient has. Additionally, a VAS score (VAS 0–100 mm) is measured during activities of daily living, with 0 mm being no pain and 100 mm the worst pain imaginable.
4. Total AOFAS score at the other time points than the primary one (at 6, 12 and 52 weeks as well as 5 years).
5. Ankle Activity Score (0–10 points) is scored according to chart based on the performable activity level
6. Subjective patient satisfaction (four categories): poor, fair, good, excellent.
7. Short Form Health Survey-36 is a health-related quality of life score (0–100 points). The higher the patient scores, the higher the disability.
8. The Global Attainment Scaling is a method of scoring based on achievement related to predetermined goals in agreement with the patient. Points are subtracted for not achieving the predefined goals or vice versa. Scores range from 100 (high functioning) to 0 (severely impaired)
9. EuroQol-5 dimensions-3 levels (EQ-5D-3L) utility score allows a patient's health to be defined by a 5-digit number.
10. PRODISQ will be used to determine indirect costs and direct costs cost-effectivity. PRODISQ is taken at baseline and every 3 months thereafter up until 1 year. This will be done in conjunction with the EQ-5D-3L.

### Loss to follow-up

The coordinating researcher will attempt to limit loss to follow-up as much as possible by contacting every patient and being present at every patient visit. All digital questionnaires will be constantly monitored to ensure they are being filled in and otherwise followed up by the coordinating researcher. In the event of patient withdrawal, an analysis of demographic and prognostic characteristics will be done on these cases and the remaining patients. As previously described by Järvinen *et al*, we will document the patient eligible for and compliant with each follow-up.[24]

### Missing items

Missing items of a score will be handled according to the instructions of the specific scales. In the event of no instructions, we will calculate the percentage of missing items on a scale. Due to the potential impact on trial conclusions, multiple imputation (if >10% missing items on a scale) will be applied.

### Sample size

Based on previous and ongoing studies, the study protocol of the RCT is designed to detect a difference of 12 points (0–100) on the AOFAS score. There is no official agreement on the minimal clinical important difference for the AOFAS score regarding ankle OA. However in relatable musculoskeletal literature, 10%–15% of the used scale was reported.[13 25 26] Our predefined minimal

clinical important difference of 12% is located within this range.[13 25 26] Based on a previous placebo-controlled RCT on injection therapy (hyaluronic acid) in ankle OA by De Groot *et al*, an SD of 16.3 can be expected.[6] Taking into account a two-sided level of significance of 5%, a power of 90% and a dropout rate of 10%, approximately 50 (45 plus 10% drop out) patients per group will be needed (n=100 in total).

### Data management
After giving permission for participating in this study, patients will receive a link to fill in digital surveys. All data gained outside Castor EDC will be stored on the AMC secured hard drive. All data will be coded and stored in the Castor EDC online database, which meets the AMC safety criteria and GCP guidelines. The primary investigator and project leader will safeguard the coded data through password secured access. All patient's data will be archived for at least 15 years and handled with in accordance with the Dutch Personal Data Protection Act (Wbp). Data protection is provided through the safety protocol of Castor EDC with automated backups and Secure Sockets Layer (SSL) security.

### Statistical analysis
A standard operating procedure will be available to logically recode and clean the data. The data will be interpreted according to a blinded data interpretation scheme described by Järvinen *et al*.[24] A statistical expert (SB) is present among the authors. The authors will interpret the statistical results until a consensus is reached. Once the authors are in agreement, the two groups will be unblinded and no changes will be made to the interpretation of the results.

### Baseline characteristics
Baseline characteristics will be analysed between groups using descriptive statistics.

### Primary outcome measure
Analysis will be performed using an intention-to-treat approach. To test for the effect of treatment on the between-group difference in primary outcome, we will use a repeated measurement general linear model. Changes from baseline to all follow-up time points will be included in the model. Adjustments will be made for those baseline variables that influenced the primary outcome with $p<0.10$.

### Secondary outcome measures
To test for the effect of treatment on between-group differences in secondary outcomes, we will use the repeated measurement general linear model. Changes from baseline to all follow-up time points will be included in the model.

### Economic analysis
In the event of a positive significant outcome, an economic analysis is needed to support a possible change of practice.

An economic analysis (costs) will be performed in order to determine cost-effectiveness.

We will assess the differences in mean quality-adjusted life years (QALYs), costs and net benefits between the PRP injection group and the placebo group using linear models. We express the cost-effectiveness by using cost-effectiveness acceptability curves from both a healthcare perspective and a societal perspective. With multiple bootstrap replicates of the average difference in costs and effects in the incremental cost-effectiveness plane, we will express the uncertainty of our cost-effectiveness analysis.

The cost-effectivity analysis will be performed with a 1 year time horizon. We use the three-level EQ-5D questionnaire (Euroqol, Rotterdam, the Netherlands) to calculate QALYs as the area under the curve of the utility scores measured over 12 months, according to the Dutch pricing system. The analysis will be based on indirect costs and direct costs and will be determined using PRODISQ. PRODISQ is taken at baseline and every 3 months thereafter up until 1 year.

### Monitoring committee
The PRIMA trial will be monitored by the clinical research unit (CRU) of the coordination study centre. The CRU aims to improve the quality of clinical research and ultimately, patient care. Before the PRIMA trial commenced, a monitoring plan was set-up. This monitoring plan facilitates compliance with the Human Research Act (WMO), GCP (ICH-GCP) guidelines (5.18.1) and/ or ISO14155, which require monitors to verify that:
1. the rights and well-being of human subjects are protected;
2. recorded study data are accurate, complete and verifiable with the source documents;
3. the conduct of the study is compliant with the currently approved protocol and with applicable laws regulatory requirements, for example, WMO, ICH-GCP, ISO14155.

Throughout the study, monitoring will be performed at initiation of the trial, at each participating centre once approximately three patients have been enrolled, after approximately 10–15 enrolled patients have completed the 26-week follow-up, after 70 patients have been enrolled and after database lock.

### Ethics and dissemination
In the events of amendments or other changes, co-researchers will be notified. If relevant, patients participating will also be notified. Intellectual data will be the property of the PRIMA study group. Participant data that underline the results reported in this article following de-identification will be shared anonymously on request following publication. Data will be shared, wherever legally and ethically possible and in line with ICMJE guidelines, with researchers who provide a methodologically sound proposal. Data will be stored in a repository (Figshare) under management of the medical library of the AMC. Any proposal should contain: title, background,

rationale, objective, methods (eligibility criteria, variables and timeframe of interest, statistical plan), information regarding publication including authorship.

The results of this project study and new knowledge will be disseminated through the Dutch Arthritis Association (ReumaNederland), presentations, news publications, blogs, websites, social media and professional organisations (rheumatology, orthopaedics, primary care medicine, sports medicine, public health) and the Dutch patient federation (Patiëntenfederatie Nederland). The patients will also be informed about the results of the study. The study will be conducted according to the principles of the Declaration of Helsinki, GCP and the Medical Research Involving Human Subjects Act (WMO). The board of each participating hospital reviewed and approved the local feasibility. Monitoring and auditing will be carried out throughout the study. In the event of serious adverse events, these will be documented and reported to the monitor and the Medical Ethics board. Eligible patients will be informed of the study and will sign an informed consent form before participating (acquired by researchers associated with the study). Provisions for post-trial care as compensation for those who suffer harm from trial participation is documented in the patient information.

## DISCUSSION

This is the first study evaluating the efficacy and cost-effectiveness of PRP injections in the treatment of ankle OA with a multicentre, stratified, block-randomised, double-blind, placebo-controlled design. The primary outcome is the AOFAS score at 26 weeks. Our study will also evaluate the long-term efficacy of PRP for up to 5 years.

PRP has been shown to alleviate symptoms of patients suffering from knee OA.[11] In the absence of effective evidence-based non-surgical interventions for ankle OA, PRP would fulfil a clear clinical need. Previous trials concerned with the efficacy of PRP for multiple indications are usually of low quality and adhere to a follow-up of 26 weeks or shorter. Furthermore, a cost-effectiveness analysis has never been performed in previous trials on PRP in the treatment of OA.

A positive study outcome would have a significant impact leading to change in clinical practice, where PRP could be offered as a non-surgical intervention. A negative outcome will prevent the widespread use of a non-efficacious treatment.

This study has potential limitations. First of all, due to financial constraints (semi-quantitative), MRI-defined cartilage degenerative changes will not be used as a secondary outcome score. Furthermore, the composition of PRP is not analysed. This decision is based on the fact that PRP preparation and composition has frequently been analysed and is typically not performed in clinical practice prior to injection.[27] The PRP we use (Hettich Rotofix32 A centrifuge and Arthrex syringes) is one out of a multitude of commercially available PRP products.

All these products differ in their production method and content, and different treatment regimens regarding dose, timing and number of injections are used in clinical practice. The generalisability to other PRP products and treatment regimens remains unknown.

## CONCLUSION

This will be the first RCT to assess the efficacy and cost-effectiveness of PRP in ankle OA.

**Author affiliations**
[1]Orthopaedic Surgery, Amsterdam UMC—Location AMC, Amsterdam, The Netherlands
[2]Amsterdam Collaboration for Health and Safety in Sports (ACHSS), AMC/VUMC IOC Research Center, Amsterdam, The Netherlands
[3]Department of General Practice, Erasmus University Medical Centre, Rotterdam, The Netherlands
[4]Orthopaedic Surgery, Erasmus Medical Center, Rotterdam, The Netherlands
[5]Sports Medicine, OLVG, The Sport Physician Group, Amsterdam, The Netherlands
[6]Academic Center for Evidence-based Sports medicine (ACES), Amsterdam UMC, Amsterdam, The Netherlands
[7]Orthopaedic Surgery, Flevoziekenhuis, Almere, The Netherlands
[8]Radiology, Amsterdam UMC—Location AMC, Amsterdam, The Netherlands
[9]Sports Medicine, Bergman Clinics, Naarden, The Netherlands
[10]OLVG, The Sport Physician Group, Amsterdam, The Netherlands
[11]Orthopaedics and Sports Medicine, Erasmus University Medical Centre, Rotterdam, The Netherlands
[12]Sports Medicine and Exercise Clinic Haarlem (SBK), Haarlem, The Netherlands
[13]Aspetar Orthopaedic and Sports Medicine Hospital, Doha, Qatar

**Contributors** All authors have been actively involved in creating and writing the study protocol, and all authors critically commented on the paper and gave their final approval of the version that the authors submitted. Authorship was and for future publications will be determined according to the ICMJE authorship recommendations. Substantial contributions to the conception or design of the work; the acquisition, analysis or interpretation of data for the work: LDAP, SMAB-Z, SG, GMMJK, RK, MM, MHM, GR, SASS, RJdV, AW and JLTol. Drafting the work or revising it critically for important intellectual content: LDAP, SMAB-Z, GMMJK, GR, RJdV, AW and JLTol. Final approval of the version to be published: LDAP, SMAB-Z, SG, GMMJ, RK, MM, MHM, GR, SASS, RJdV, AW and JLTol. Agreement to be accountable for all aspects of the work in ensuring that questions related to the accuracy or integrity of any part of the work are appropriately investigated and resolved: LDAP, SMAB-Z, SG, GMMJK, RK, MM, MHM, GR, SASS, RJdV, AW and JLTol.

**Funding** This multicentre randomised controlled trial was supported by a grant from the Dutch Arthritis Foundation (ReumaNederland).

**Competing interests** None declared.

**Patient consent for publication** Not required.

**Ethics approval** The study is approved by the local Medical Ethics Review Committee Amsterdam Medical Center, Amsterdam, the Netherlands (ABR 2018–042, approved 23-7-2018) in accordance with the WHO Data Set.

**Provenance and peer review** Not commissioned; externally peer reviewed.

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
