## [Reviewer comments · BMJ Open]

ARTICLE DETAILS

TITLE (PROVISIONAL)	Platelet Rich plasma Injection Management for Ankle osteoarthritis study (PRIMA) Protocol of a Dutch multi-center, stratified, block-randomized, double-blind, placebo-controlled trial
AUTHORS	Paget, Liam David Andrew; Bierma-Zeinstra, Sita M.A.; Goedegebuure, S; Kerkhoffs, Gino M.M.J.; Krips, R; Maas, Mario; Moen, Maarten H; Reurink, G; Stufkens, Sjoerd A.S.: de Vos, Robert-Jan; Weir, A; Tol, J L

VERSION 1 – REVIEW

REVIEWER	Jun Liu The Second Affiliated Hospital of Guangzhou University of Chinese Medicine, Guangzhou, China.
REVIEW RETURNED	11-Jun-2019

GENERAL COMMENTS	The protocol is well designed. This paper has a potential to be accepted. Can you discuss the mechanisms for PRP injections in the treatment of ankle OA? In addition, is there a statistical expert involved in your clinical trial?
---

VERSION 1 – AUTHOR RESPONSE

Reviewer: The protocol is well designed. This paper has a potential to be accepted.

Response to reviewer: We thank the reviewer for his/her kind words.

Action taken: None required

Reviewer: Can you discuss the mechanisms for PRP injections in the treatment of ankle OA?

Response to reviewer: We thank the reviewer for this point and have added a paragraph discussing the mechanisms of PRP.

Action taken: page 5 line 100-109:

“Platelet Rich Plasma (PRP) is defined as plasma containing a concentration of at least 1,000,000 platelets/ μ l.⁴ Growth factors are stored in α -granules within platelets, and are released in a selective manner upon activation. Growth factors released from the α -granules of platelets are assumed to provide the regenerative benefits of PRP.

Recent reviews concluded that in animal models PRP can diminish multiple inflammatory IL-1 mediated effects.⁵ Due to this local anti-inflammatory response, PRP might have an indirect analgesic effect. The second suggested effect might be protection of the cartilage from destructive pro-inflammatory interleukines. This is mediated due to an increased mRNA expression of proteoglycan core protein in the articular cartilage and decreased chondrocyte apoptosis.⁵ Consequently, PRP could positively influence the collagen network of the cartilage.”

Reviewer: In addition, is there a statistical expert involved in your clinical trial?

Response to reviewer: We do indeed have a statistical expert and have added this to the protocol accordingly.

Action taken: page 18, line 364: "A statistical expert (SB) is present among the authors."